# INSCAL: CALIBRATED MULTI-SOURCE FULLY TEST-TIME PROMPT TUNING FOR OBJECT DETECTION

## ABSTRACT

Test-time prompt tuning (TPT) has emerged as a powerful technique for adapting pre-trained vision-language models (VLMs) to diverse downstream tasks, including image classification and visual reasoning. With the rise of text-driven object detectors, we extend TPT to object detection, unlocking new capabilities for cross-domain adaptation. However, a critical challenge in TPT is the inherent miscalibration caused by entropy minimization: domain shifts often lead to incorrect predictions, and enforcing high confidence exacerbates miscalibration, ultimately degrading performance. To tackle this, we introduce InsCal, a novel framework designed to enhance cross-domain object detection through three key innovations: (1) extending TPT to a multi-source paradigm, enabling knowledge aggregation across diverse domains; (2) reducing domain gaps via a novel text-driven style transfer strategy that aligns features to the source domain without requiring reference images; and (3) refining the entropy minimization objective with instance-specific calibration, ensuring robust and well-calibrated adaptation. Our approach not only mitigates miscalibration but also significantly improves cross-domain object detection performance, setting a new benchmark for test-time adaptation in VLMs.

## 1 INTRODUCTION

By encoding a wide range of visual concepts after training on millions of noisy image-text pairs, pre-trained vision-language models (VLMs) have shown great promise for the development of foundational models applicable to various downstream vision tasks Radford et al. (2021); Zhou et al. (2022b). Built upon VLMs' joint embedding space of images and text, text-driven object detectors aim to detect objects that go beyond predefined categories by leveraging large-scale image-text datasets. They frame open-vocabulary object detection as a task of image-text matching, allowing the model to recognize and locate objects that may not have been explicitly included in the training categories Zareian et al. (2021); Phoo & Hariharan (2022); Yao et al. (2022); Feng et al. (2022); Liu et al. (2024a); Yao et al. (2023).

Despite the remarkable generalization ability from base classes to novel classes, the performance of text-driven object detectors suffers when the target domain displays drastically different distributions. For example, GDINO Liu et al. (2024a) is the latest transformer-based object detection with large scale grounded pre-training for zero-shot transfer. As shown in Figure 1a, we tested the cross-domain performance using pre-trained GDINO model on the Diverse Weather Dataset (DWD) dataset Wu & Deng (2022). DWD is a semantic urban scene understanding dataset designed to capture urban environments under a variety of weather and time conditions. DWD contains five distinct domains, each representing a different combination of weather and time conditions: DayClear, NightClear, NightRainy, DuskRainy and DayFoggy. The zero-shot performance is obtained by using pre-trained GDINO without any adaptation. The fine-tune performance is obtained with fine-tuning pre-trained GDINO models on corresponding datasets. From Figure 1a, we observed a noticeable performance gap between the fine-tune and zero-shot transfer of GDINO. Especially in NightRainy and DuskRainy domain, GDINO fails to give proper predictions. This degradation in average precision (AP) when using zero-shot transfer highlights the limitations of directly applying pre-trained object detectors on out-of-domain data. The results illustrate that without fine-tuning, pre-trained models struggle to generalize effectively to new, unseen domains, leading to less accurate predictions and overall reduced performance.

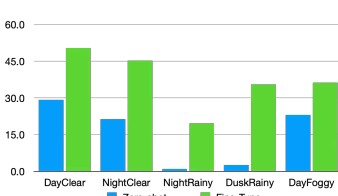 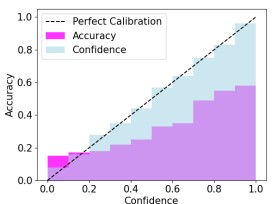 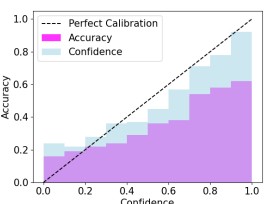

(a) Cross-domain performance (mAP) of GDINO on DWD dataset w/ and w/o fine-tuning.

(b) Cross-domain performance using TPT w/o calibration (D-ECE: 19.5%)

(c) Cross-domain performance using TPT w/ calibration (D-ECE: 13.2%)

Figure 1: Experimental Illustrations.

Test-time adaptation (TTA) aims to adapt a pre-trained model during testing under distribution shifts Wang et al. (2020); Liang et al. (2020); Yang et al. (2021); Karani et al. (2021); Wang et al. (2021c); Liu et al. (2024b). Only a few previous works have leveraged TTA for object detection Chen et al. (2023); Ruan & Tang (2024); Cao et al. (2024). However, these methods can not generalize well to text-driven object detectors. In this work, we explore TTA for text-driven object detectors with test-time prompt tuning (TPT). Prompt tuning proposes to directly learn prompts using training data from downstream tasks by treating prompt embeddings as trainable parameters differentiate with respect to the loss function, which requires training data with annotations Du et al. (2022); Zhou et al. (2022a). Test-time prompt tuning (TPT) address this problem by tuning the prompt on the fly using only the given test sample Shu et al. (2022). The tuned prompt is adapted to each task by minimizing the entropy of the top confident samples which are obtained using different augmented views, making it suitable for zero-shot generalization without requiring any task-specific training data or annotations. Subsequent works such as DART Liu et al. (2024b), DiffTPT Feng et al. (2023) build on the entropy minimization scheme and utilize techniques such as incorporating image prompt or data generation using diffusion models. However, this line of work poses a potential risk of over-trust on the model, that is, generating incorrect predictions with high confidence Ma et al. (2024). In Figure 1b, we conduct experiment on cross domain dataset (DayClear to NightClear) with TPT. The huge gap between the output confidence and the actual accuracy in in the left figure of Section 1 shows that directly applying TPT on cross-domain task lead to overconfident results. In the right figure, we show that after applying our proposed calibrated learning objective, the miscalibration issue is greatly reduced.

In this work, we propose the instance-specific calibrated test-time prompt tuning for object detection (InsCal), designed toward addressing the risk of miscalibration during test-time adaptation. To our best knowledge, model calibration poses a novel challenge in object detection that has not been addressed by any existing work due to the potential domain shift coupled with the lack of labeled target samples. To achieve reliable object detection when deploying a model to a new test domain with potential domain gap and no label information, InsCal integrates three key innovations: First, we extend Test-Time Prompt Tuning (TPT) to a multi-source setting, enabling the model to leverage knowledge from multiple pre-trained source models, thereby enhancing its robustness across diverse domains. Second, we introduce text-guided image augmentation, a technique aimed at explicitly reducing the domain gap between source and target domains, which helps to mitigate performance degradation caused by domain shifts. Finally, we propose a calibrated entropy minimization objective, which incorporates a calibration factor based on the largest and second-largest logits for each instance, effectively addressing the issue of overconfidence in predictions and improving the model's reliability during test-time adaptation, which is essential for many critical domains (e.g., autonomous driving and military operations).

We conduct experiments with fully test-time adaptation on cross-domain object detection datasets. InsCal reduces the expected calibration error (D-ECE) around 10%. The contributions of this paper is summarized as follows: (1) We investigate that large pre-trained object detectors suffer from performance degradation for fully test-time adaptation (FTTA). Test-time Prompt Tuning (TPT) also suffers from miscalibration due to overconfidence. (2) We propose a principled method that seamlessly integrates multiple source models, effectively bridging semantic gaps by text-guide feature augmentation. Additionally, we design a calibrated entropy minimization technique to address miscalibration, ensuring more accurate test-time adaptation for object detection. (3) Experiments

conducted on multiple cross-domain object detection datasets verify that our method effectively reduce domain gaps and miscalibration.

## 2 RELATED WORKS

**Test-time Adaptation**    Test-time Adaptation (TTA) aims to adapt model weights pre-trained on the source domain to a unseen domain. During adaptation, TTA only has access to the pre-trained models and unlabeled target data. TTA can be categorized into test-time (source-free) domain adaptation (SFDA), test-time batch adaptation (TTBA), online test-time adaptation (OTTA) and fully test-time adaptation (FTTA) Liang et al. (2024). SFDA Liang et al. (2020); Yang et al. (2021); Tian et al. (2021); Nayak et al. (2021); Liang et al. (2021) is able to utilize all unlabeled test data from the target domain during a multi-round adaptation before generating final predictions. TTBA Schneider et al. (2020); Sun et al. (2020); Park et al. (2020); Karani et al. (2021); Wang et al. (2021c) only has access to one or a few instances (a batch) during this process. For OTTA Ioffe (2015); Wang et al. (2020); Boudiaf et al. (2022), the adaptation is conducted in an online manner, where where each batch is only observed once. FTTA Shu et al. (2022); Liu et al. (2024b); Ruan & Tang (2024) adapts the pre-trained model on-the-fly with a single test sample. Test-time adaptation for object detection is a relatively under-explored field. STFAR Chen et al. (2023) generates pseudo labels via a regularized feature alignment self-training paradigm for the adaptation of source object detector. CTAOD Cao et al. (2024) addresses continual test-time adaptation (CTTA) where the target domain distribution undergoes temporal changes with object-level contrastive learning, dynamical skips and stochastic restoration. IOUFilter Ruan & Tang (2024) studies fully test-time adaptation which adapts pre-trained source detectors with only a single test-image by acquiring high-quality pseudo labels. In this work, we mainly focus on the application of FTTA on text-drive object detectors.

**Test-Time Prompt Tuning**    Test-time prompt tuning (TPT) provides a solution for FTTA on pre-trained vision-language models (VLMs) via learnable prompts. TPT is first proposed to address image classification and visual reasoning by Shu et al. (2022), which aims to learn text prompts using an entropy minimization objective with consistency constraints across different augmented views of the single test image. DART Liu et al. (2024b) extends TPT by further incorporating the learning of image prompt during test-time. Instead of using traditional augmentation techniques, such as random cropping, or translation, DiffTPT Feng et al. (2023) leverages pre-trained diffusion models to generate augmented views. PromptAlign Samadh et al. (2023) handles domain shift explicitly minimizing the feature distribution shift. SwapPrompt Ma et al. (2024) employs a framework with an online prompt and a target prompt to better retain historical information. VPA Sun et al. (2023) focus on generalizing visual prompting with test-time adaptation. UPT He et al. (2023a) adopts a mean-teacher mechanism to learn text-prompt in a zero-shot manner for object detection tasks. While effective, UPT only utilize a single source model trained from a single source domain, struggles with diverse unknown target domains. In this work, we aim to address the overconfidence issue induced by the entropy minimization objective in test-time prompt tuning. We first extend test-time prompt tuning with multiple pre-trained source models to integrate knowledge from different domains; to reduce domain gaps, we propose text-guide image generation to generate augmented views with source domain styles; we then design a calibrated entropy minimization objective for the calibrating the instance specific weights.

## 3 PRELIMINARIES

**Calibration for object detection.**    Given a dataset $\mathcal{D} = \{(\mathbf{x}_i, y_i, \mathbf{b})\}_{i=1}^N$, where $\mathbf{x}_i \in \mathbb{R}^{H \times W \times C}$ is the $i$-th image, and $y_i \in \{1, ... K\}$ is the corresponding ground truth label, where $K$ denotes the number of classes, $H$, $W$ and $C$ are the width, height, and number of channels of the image. $\mathbf{b}_i \in [0, 1]^4$ denotes the bounding box annotation. Given the predicted object label $\hat{y}$ and the predicted object location $\hat{\mathbf{b}}$ with a confidence score $\hat{s}_{\text{conf}}$, a perfect calibration of a object detector is defined as Kuppers et al. (2020)

$$P(\hat{y} = y, \hat{\mathbf{b}} = \mathbf{b}, \hat{s}_{\text{conf}} = s_{\text{conf}}) = s_{\text{conf}} \ \ \forall s_{\text{conf}} \in [0, 1] \quad (1)$$

where $P(\hat{y} = y, \hat{\mathbf{b}} = \mathbf{b}, \hat{s}_{\text{conf}} = s_{\text{conf}})$ is the prediction performance with a confidence score $s_{\text{conf}}$, indicating that the object class is correctly labeled $\hat{y} = y$ and the intersection-over-union (IOU) is larger than a predefined threshold $\gamma$ $IoU(\hat{\mathbf{b}}, \mathbf{b}) > \gamma$.

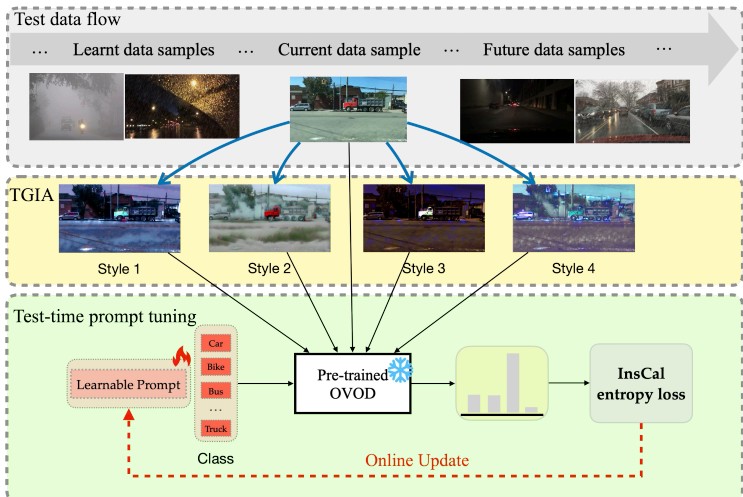

Figure 2: Overall Framework: Given a streamline of test examples, each test image is augmented with source domain styles using Text-Guide Image Augmentation (TGIA). Multi-source Test-time Prompt Tuning (MSTPT) extracts TGIA image features with multiple source image encoders. Given the prompted text features, InsCal outputs prediction probability for each augmentation. The InsCal entropy loss is computed by filtering out high entropy predictions and assigning proper instance-specific calibration weights. The InsCal entropy loss is then back-propagated to update the prompt.

The quantification the miscalibration is measured by the detection expectation of calibration error (D-ECE) Kuppers et al. (2020):

$$\mathbb{E}[|P(\hat{y} = y, \hat{\mathbf{b}} = \mathbf{b}, \hat{s}_{\text{conf}} = s_{\text{conf}}) - s_{\text{conf}}|] \tag{2}$$

To approximate D-ECE, the continuous space of the confidence $\hat{s}_{\text{conf}}$, and the box property space in each dimension are equally divided into $M$ bins, and

$$\text{D-ECE} = \sum_{m=1}^{M} \frac{|I(m)|}{|\mathcal{D}|} |\text{prec}(m) - \text{conf}(m)| \tag{3}$$

where $I(m)$ is the set of all samples in a single bin, $|\mathcal{D}|$ is the number of samples, prec(m) and conf(m) denote the average precision and confidence in each bin, respectively.

## 4 METHODOLOGY

**Overview.** The overall pipeline of InsCal is illustrated in Figure 2. Given textual style descriptions of each domain, the target image is augmented through TGIA, and the resulting views are used to construct an instance-specific calibrated entropy. This entropy guides the update of the learnable prompts, effectively mitigating the overconfidence issue.

**Problem definition.** We consider the multi-source test-time prompt-tuning setting. Given $S$ source model $f_\theta^s$, each pre-trained on a different source domain $\mathcal{D}_s$, where each domain s is accompanied by a short text description of its style domain$_{sty}^s$. Each source model $f_\theta^s$ is explicitly represented as an image encoders $\text{ENC}_I^s$ and the text encoder $\text{ENC}_T$. At test time, given a single target-domain image $\mathbf{x}_{\text{test}} \in \mathcal{D}_T$, our objective is to learn an optimal prompt $\mathbf{p}^*$ that maximizes adaptation performance to the target domain $\mathcal{D}_T$.

### 4.1 TEXT-GUIDE IMAGE AUGMENTATION

As shown in the left of Figure 3, given a test image $\mathbf{x}_{\text{test}}$, a target style text tgt$_{sty}$ and a source style text src$_{sty}$, TGIA $\mathcal{A}_\theta(\cdot)$ generates an augmented view $\mathcal{A}_\theta(\mathbf{z})$ of the target image in the corresponding

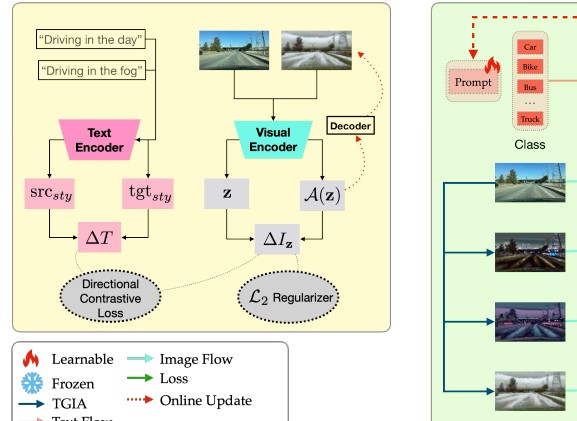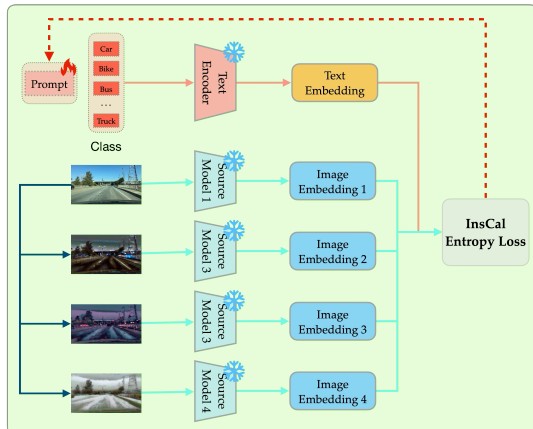

Figure 3: **Left: Overview of TGIA.** TGIA leverages textual descriptions of source and target domain styles to transfer a target image into the style of the source domain. **Right: Overview of InsCal.** InsCal extends TPT to the multi-source setting by integrating multiple source models and addressing miscalibration through an instance-specific calibration entropy loss.

source style by minimizing the following regularized directional contrastive loss:

$$
\theta^* = \min_\theta \sum_{\mathbf{z}} 1 - \frac{|\Delta I_{\mathbf{z}}|}{|\Delta T|} \cdot \frac{\Delta I_{\mathbf{z}} \cdot \Delta T}{|\Delta I_{\mathbf{z}}||\Delta T|} + \lambda \|\mathcal{A}_\theta(\mathbf{z}) - \mathbf{z}\|_2^2,
$$
$$
\Delta I_{\mathbf{z}} = \text{ENC}_I(\mathcal{A}_\theta(\mathbf{z})) - \text{ENC}_I(\mathbf{z}), \tag{4}
$$
$$
\Delta T = \text{ENC}_T(\text{tgt}_{sty}) - \text{ENC}_T(\text{src}_{sty}),
$$

where $\mathbf{z} = \text{ENC}_I(\mathbf{x}_{\text{test}}^{\text{crop}})$ is the image embedding of a patch obtained by randomly taking multiple crops from the test image $\mathbf{x}_{\text{test}}$. The first term aims to align the direction of the image style transformation (induced by TGIA) with the textual style transformation (from target style to source style) in the latent space. $\Delta I_{\mathbf{z}}$ represents the change in the image embedding due to TGIA applied to the test image. It's computed as the difference between the embeddings of the augmented image, $\mathcal{A}_\theta(\mathbf{z})$, and the original image patch embedding, $\mathbf{z}$. $\Delta T$ represents the direction of the style transformation from the target to the source, based on the text encodings of the target style $\text{tgt}_{sty}$ and source style $\text{src}_{sty}$. By minimizing the cosine similarity between $\Delta I_{\mathbf{z}}$ and $\Delta T$, the loss encourages $\Delta I_{\mathbf{z}}$ (the change in image style) to align with $\Delta T$ (the intended style direction in text). This alignment effectively guides the test image's style toward the source domain style described in text. The magnitude scaling factor $\frac{|\Delta I_{\mathbf{z}}|}{|\Delta T|}$ encourages the augmentation's transformation magnitude to closely match that of the desired text-guided shift, making alignment stronger. The second term is a $\mathcal{L}_2$ regularization encourages the augmented image $\mathcal{A}_\theta(\mathbf{z})$ to remain close to the original image patch $\mathbf{z}$ in terms of content. This term enforces content similarity in feature space, allowing flexibility in low-level style features while keeping the main content of the test image. $\lambda$ is a hyperparameter that controls the relative importance of the perceptual content preservation.

The textual description of the domain style is a straightforward sentence summarizing the overall style of the dataset. For example, for dataset Watercolor2k Inoue et al. (2018), the textual description is "a drawing in watercolor style". TGIA does not require access to the source data and adheres to the FTTA requirement because it relies solely on a high-level textual description of the source domain style, rather than any specific source images. For simplicity, we use $\mathcal{A}(\cdot)$ instead of $\mathcal{A}_\theta(\cdot)$ in the following sections.

## 4.2 MULTI-SOURCE TEST-TIME PROMPT TUNING

In the multi-source test-time prompt-tuning (MSTPT) setting, we are provided with $S$ pre-trained source models. In this work, we adopt GDINO Liu et al. (2024a) as the base model. GDINO is a text-driven object detector pre-trained on large-scale datasets. To obtain multiple source models, we fine-tune GDINO on datasets $\{\mathcal{D}_s\}_{s=1}^S$ from different source domains, , resulting in a set of

source image encoders $\{\text{ENC}_I^s\}_{s=1}^S$. Since the semantic representation of text (e.g., object labels, descriptions) remains relatively domain-agnostic, we use the same text encoder $\text{ENC}_T$ together with all source image encoders. We then generate augmented views of the test image $\{\{\mathcal{A}_j^s(\mathbf{x}_{\text{test}})\}_{j=1}^N\}_{s=1}^S$ using TGIA given the source style description, where $N$ augmented views are obtained for each source domain.

Built upon the pre-trained source models and the augmented views, we propose enhancing the calibration of test-time prompt-tuning for object detection in three key ways: (1) integrating information from multiple sources to fully leverage the knowledge of multiple pre-trained source models; (2) explicitly reducing domain gaps between source models and target images to achieve highly accurate, confident predictions; and (3) introducing a novel calibrated objective to overcome overconfidence in entropy minimization. As shown on the right of Figure 3, both the text encoder and image encoders remain frozen during adaptation, while the augmented views in each source style are passed through their corresponding image encoder. To encourage consistency, the prompt prompt $\mathbf{p} \in \mathbb{R}^{L \times D}$ is optimized in the text embedding space by minimizing the entropy of the averaged prediction distribution across all $S \times N$ augmented views, where $L$ is the number of tokens, and $D$ is the embedding size.

$$\mathbf{p}^* = \min_{\mathbf{p}} - \sum_{i=1}^K \tilde{p}_{\mathbf{p}}(y_i|\mathbf{x}_{\text{test}}) \log \tilde{p}_{\mathbf{p}}(y_i|\mathbf{x}_{\text{test}}) \tag{5}$$

$$\tilde{p}_{\mathbf{p}}(y_i|\mathbf{x}_{\text{test}}) = \frac{1}{SN} \sum_{s=1}^S \sum_{j=1}^N p_{\mathbf{p}}(y_i|\mathcal{A}_j^s(\mathbf{x}_{\text{test}})) \tag{6}$$

$$p_{\mathbf{p}}(y_i|\mathcal{A}_j^s(\mathbf{x}_{\text{test}})) = \frac{\exp(SIM_i/\tau)}{\sum_{k=1}^K \exp(SIM_k/\tau)} \tag{7}$$

where $p_{\mathbf{p}}(y_i|\mathcal{A}_j^s(\mathbf{x}_{\text{test}}))$ is the vector of class probabilities produced by the $s$-th source model when provided with prompt $\mathbf{p}$ and the $j$-th augmented view with $s$-th source style of the test image. $SIM_i = \cos(\text{ENC}_I^s(\mathcal{A}_j^s(\mathbf{x}_{\text{test}})), \text{ENC}_T(\mathbf{p}_i))$ is the cosine similarity between the prompted text feature $\text{ENC}_T(\mathbf{p}_i)$ and the augmented image feature of $j$-th view of $s$-th source image encoder $\text{ENC}_I^s(\mathcal{A}_j^s(\mathbf{x}_{\text{test}}))$. Given a confidence selection threshold $\sigma$, we filter out views with high entropy prediction in each source $s$:

$$\tilde{p}_{\mathbf{p}}(y_i|\mathbf{x}_{\text{test}}) = \frac{1}{\rho SN} \sum_{s=1}^S \sum_{j=1}^N \mathbb{1}[\mathbf{H}(p_i) \leq \sigma] p_{\mathbf{p}}(y_i|\mathcal{A}_j^s(\mathbf{x}_{\text{test}})) \tag{8}$$

where $\rho$ is the cutoff percentile on $SN$ total views, $\mathbb{1}[\cdot]$ is a indicator function which assigns 1 when $\mathbf{H}(p_i) \leq \sigma$ and 0 otherwise. $\mathbf{H}(p_i)$ measures the self-entropy of the prediction on an augmented view.

### 4.3 CALIBRATED ENTROPY MINIMIZATION

A key drawback of minimizing average prediction entropy is that it promotes high-confidence (low-entropy) predictions across all augmentations, even for incorrect ones, leading to overly confident results Tao et al. (2023); Tan et al. (2024); Yang et al. (2024). To reduce overconfidence in entropy minimization while preserving the benefits of enhanced prediction precision, we propose calibrated test-time prompt tuning leveraging the highest-ranked prediction along with the next best prediction. The probability of class $y_i$ among $K$ classes is denoted as $p_i = p_{\mathbf{p}}(y_i|\mathcal{A}_j^s(\mathbf{x}_{\text{test}}))$. We further define $p^{\text{1st}} = p_{\mathbf{p}}(y^{\text{1st}}|\mathcal{A}_j^s(\mathbf{x}_{\text{test}}))$ as the highest prediction and $p^{\text{2nd}} = p_{\mathbf{p}}(y^{\text{2nd}}|\mathcal{A}_j^s(\mathbf{x}_{\text{test}}))$ as the second highest prediction following $p^{\text{1st}}$. Based on these definitions, the calibrated multi-source test-time prompt-tuning objective is formulated as

$$\mathbf{p}^* = \min_{\mathbf{p}} \frac{1}{SN} \sum_{i=1}^K \sum_{s=1}^S \sum_{j=1}^N \tilde{H}[\tilde{p}_{\mathbf{p}}(y_i|\mathbf{x}_{\text{test}})] \tag{9}$$

$$\tilde{H}[\tilde{p}_{\mathbf{p}}(y_i|\mathbf{x}_{\text{test}})] = -(1 + (p^{\text{1st}} - p^{\text{2nd}})^\alpha) p_i \log p_i \tag{10}$$

The term $1 + (p^{\text{1st}} - p^{\text{2nd}})^\alpha$ serves as a calibration factor. It adapts to the specific confidence of the prediction, influencing how much weight is assigned to each augmented view. When $p^{\text{1st}}$ is much larger than $p^{\text{2nd}}$ (indicating high confidence), the calibration factor becomes larger. This increases the

Table 1: mAP (%) and D-ECE (%) on different sub-dataset of the Art Image Dataset. FR and UDA are pre-trained on PASCAL VOC dataset. GDINO is pre-trained on O365, GoldG, and Cap4M. FTTA methods are fine-tuned on corresponding source data with the pre-trained GDINO. More details of the baselines are presented in the Appendix.

| Domains | | Comic | | Clipart | | Watercolor | |
|---|---|---|---|---|---|---|---|
| Methods | | mAP | D-ECE | mAP | D-ECE | mAP | D-ECE |
| with access to source data | | | | | | | |
| UDA | FR | 25.0 | 18.2 | 29.8 | 16.3 | 52.0 | 17.3 |
| | UAN | 25.5 | - | 30.3 | - | 53.3 | - |
| | CMU | 30.1 | - | 32.1 | - | 53.9 | - |
| | DAF | 28.3 | - | 31.3 | - | 49.3 | - |
| | MAF | 29.3 | - | 32.2 | - | 49.2 | - |
| | HTCN | 24.0 | - | 34.7 | - | 52.1 | - |
| | CAD | 28.8 | - | 34.2 | - | 52.8 | - |
| | IDF | 24.8 | - | 32.7 | - | 52.5 | - |
| | USDAF | 32.6 | - | 38.4 | - | 55.2 | - |
| | CODE | **33.8** | 17.5 | **39.4** | 17.1 | **55.8** | 17.3 |
| target data are presented in an online manner | | | | | | | |
| FTTA | GDINO | 25.9 | 17.2 | 30.5 | 16.9 | 52.8 | 17.0 |
| | Tent | 25.5 | 16.8 | 30.3 | 16.3 | 52.5 | 16.5 |
| | TPT | 25.9 | 16.2 | 30.6 | 15.5 | 53.0 | 16.0 |
| | IOUFilter | 20.2 | 17.5 | 29.6 | 17.4 | 35.8 | 17.5 |
| | C-TPT | 28.4 | 16.9 | 32.9 | 17.2 | 49.7 | 17.3 |
| | ZS-Norm | 29.2 | 16.5 | 33.4 | 16.8 | 50.4 | 17.0 |
| | Penalty | 29.7 | 16.6 | 33.8 | 16.9 | 50.9 | 17.2 |
| | SaLS | 29.8 | 16.5 | 34.0 | 16.6 | 51.2 | 16.9 |
| | O-TPT | 30.4 | 16.2 | 34.5 | 15.6 | 51.9 | 16.1 |
| | InsCal | **34.3** | **15.4** | **39.9** | **14.7** | **56.3** | **15.2** |

importance of this confident prediction. When $p^{1st}$ and $p^{2nd}$ are close, the model is less confident, and the calibration factor reduces the importance of this prediction. This down-weights the prediction, thus preventing overconfident but inaccurate predictions. $\alpha$ is a hyperparameter that controls how strongly the model should adjust its confidence based on the difference between the top two logits. A larger $\alpha$ makes the calibration more sensitive to the difference between $p^{1st}$ and $p^{2nd}$, leading to more drastic adjustments. A smaller $\alpha$ results in more gradual adjustments.

## 5 EXPERIMENTS

**Datasets.**    Diverse Weather Dataset (DWD) Wu & Deng (2022) is a cross-domain object detection dataset focuses on semantic understanding of urban street scenes with instance-level annotations. DWD consists of five domains: Daytime Clear, Daytime Foggy, Dusk Rainy, Night Rainy and Night Clear. Each domains collects urban street scenes dataset with a specific weather conditions (i.e., clear, foggy, or rainy) at a time (i.e., day, dusk, or night). All the datasets contain bounding box annotations within 7 classes objects: *bus, bike, car, motorbike, person, rider*, and *truck*. The dataset size for DWD is 27708, 3775, 3501, 2494, and 26158 for Daytime Clear, Daytime Foggy, Dusk Rainy, Night Rainy and Night Clear, respectively. Another cross-domain dataset we use is the Art Image dataset with different artistic styles including Clipart1k, Comic2k, and Watercolor2k Inoue et al. (2018), where Clipart1k contains 1000 clipart images, Comic2k contains 2000 comic images.

**Metrics**    Mean Average Precision with threshold 0.5 (mAP@0.5) is used to measure the performance of all experiments. mAP@0.5 considers a prediction as a true positive if it matches the ground-truth label and has an intersection over union (IOU) score of more than 0.5 with ground-truth bbox.

### 5.1 MAIN RESULTS

**Art Image Dataset**    In Table 1, we present the mAP and D-ECE results for the Art Image dataset. For each domain, we use the rest two as source. For certain baselines, we directly report the results from their respective papers, where D-ECE values are not available. Notably, InsCal surpasses UDA methods despite their advantage of accessing source data, as these methods fail to address the issue of model overconfidence. In general, FTTA baselines underperform compared to UDA methods due to the inherent limitation of lacking source data access. Calibrated TPT methods outperforms other FTTA method since they address the overconfidence issue. Our method InsCal effectively leverages knowledge from multiple source domains, achieving superior performance over UDA

approaches. Furthermore, our calibrated entropy minimization strategy significantly reduces D-ECE, demonstrating its effectiveness in improving model calibration. The detailed analysis of each class is presented in the Appendix.

**DWD Dataset.** In Table 2, we present the main results for DWD dataset, including mAP and D-ECE for each domain. We categorize the comparative baselines into UDA, SFDA, and FTTA based on source and target data availability, with the best performance in each category highlighted in bold. Our method consistently achieves the lowest D-ECE across all categories and sub-domains, highlighting that traditional UDA, SFDA, and FTTA methods suffer from severe miscalibration. While calibrated TPT methods partially alleviate this issue, our approach notably reduces D-ECE from approximately 20% to 10%, demonstrating the effectiveness of calibrated entropy minimization. In terms of mAP, InsCal outperforms competing methods in Dusk Rainy, Night Rainy, and Night Clear domains. On the Day Foggy benchmark, our method performs competitively, trailing only slightly behind two UDA methods, despite their significant advantage of full access to both source data and unlabeled target data. Additionally, in Figure 4, we observe that our method effectively aligns confidence scores with actual prediction accuracy, leading to more reliable and well-calibrated detections. The detailed analysis for each class in presented in the Appendix.

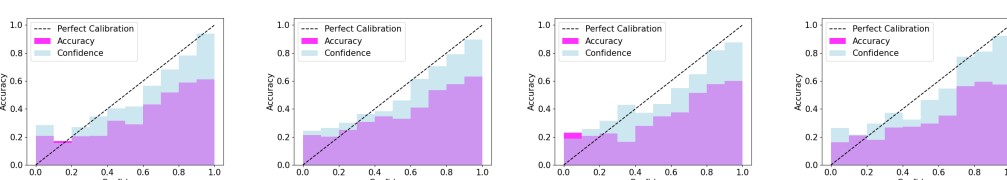

(a) Night Rainy w/o calibration. (D-ECE: 13.25%)  (b) Night Rainy w/ calibration. (D-ECE: 12.18%)  (c) Dusk Rainy w/o calibration. (D-ECE: 15.12%)  (d) Dusk Rainy w/ calibration. (D-ECE: 14.48%)

Figure 4: Multi-source TPT fine-tuned on Night Rainy and Dusk Rainy from the DWD dataset Wu & Deng (2022) w/ and w/o calibration loss in training.

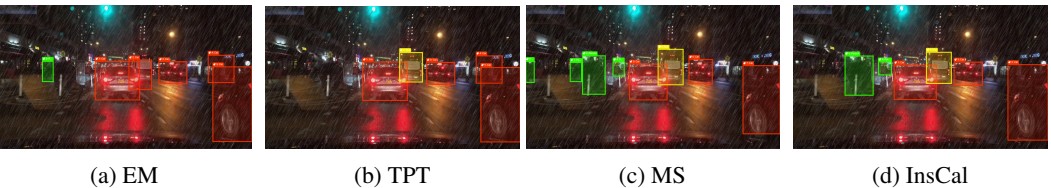

(a) EM  (b) TPT  (c) MS  (d) InsCal

Figure 5: Qualitative analysis on different components from our model to object detection performance on one image of Night Clear.

## 5.2 ABLATION STUDIES

**Effectiveness of each component.** In this ablation, we study the effectiveness of each component in InsCal. As shown in Table 3, using entropy minimization (EM) has little transferability to extremely different target domains. By using augmented views and constraining them to with low entropy, TPT improve the performance over EM by 1.6. Utilizing multi-source models during training has the advantage of aggregating information from multiple domains, thus further improve the performance. TGIA improve the performance by reducing domain gaps. And using calibrated loss improve the performance by preventing overconfidence. In Figure 5, we provide some qualitative results for InsCal. We observe that EM misclassifies multiple objects including car, bus, rider and person. TPT correctly identifies some cars and person, but misclassifies truck and bus. MS can identify more cars, but mistakenly identify some other objects as trucks. InsCal correctly identifies all the objects without mistakes.

**Extension to open vocabulary object detection.** We extend our method to open-vocabulary object detection (OVOD) on the DWD dataset. The results for the Day Foggy scenario are presented in Table 4, where the novel category Traffic Light is highlighted in gray. Our approach achieves the highest mAP across all categories except Car, where FR attains the best performance. However, FR exhibits the worst performance on the novel category, highlighting the effectiveness of our method in seamlessly adapting to OVOD. Furthermore, the lowest D-ECE score demonstrates that our approach mitigates overconfidence issues, enhancing robustness in open-vocabulary settings.

Table 2: mAP (%) and D-ECE (%) results. For each target domain, Day Clear and the rest three domains are used as the source domains for the multi-source methods. For single-source UDA and SFDA, Day Clear is used as the source following the typical setting Wu & Deng (2022); Vidit et al. (2023); Fahes et al. (2023).

| | Domain | Day Foggy | | Dusk Rainy | | Night Rainy | | Night Clear | |
|---|---|---|---|---|---|---|---|---|---|
| | Methods | mAP | D-ECE | mAP | D-ECE | mAP | D-ECE | mAP | D-ECE |
| | with access to source data | | | | | | | | |
| UDA | FR | 32.0 | - | 26.0 | - | 12.4 | - | 34.4 | - |
| | SW | 30.8 | - | 26.3 | - | 13.7 | - | 33.4 | - |
| | IBNNet | 29.6 | - | 26.1 | - | 14.3 | - | 32.1 | - |
| | IterNorm | 28.4 | - | 22.8 | - | 12.6 | - | 29.6 | - |
| | ISW | 31.8 | - | 25.9 | - | 14.1 | - | 33.2 | - |
| | SDGOD | 33.5 | 18.8 | 27.9 | 18.7 | 16.6 | 18.5 | 36.6 | 19.0 |
| | CLIPAug | 38.5 | 18.4 | **28.2** | 18.5 | 18.7 | 18.2 | 36.9 | 18.3 |
| | PODA | **38.9** | **17.5** | 27.5 | **17.9** | **19.5** | **17.7** | **37.4** | **17.8** |
| | with access to all target data | | | | | | | | |
| SFDA | SED | 29.4 | **14.2** | 21.1 | **15.4** | 15.1 | 14.6 | 33.4 | 15.5 |
| | MSMT | **36.8** | 14.5 | **32.0** | 15.6 | **16.5** | 14.6 | **37.7** | 15.7 |
| | MixUp | 31.5 | 14.8 | 30.8 | 15.7 | 15.5 | 14.5 | 35.0 | 15.6 |
| | HCL | 30.2 | 14.5 | 26.9 | **15.4** | 15.3 | 14.3 | 30.8 | 15.5 |
| | IRG | 35.2 | 15.1 | 30.5 | 15.6 | 15.8 | **14.1** | 36.7 | **15.1** |
| | target data are presented in an online manner | | | | | | | | |
| FTTA | GDINO | 34.1 | 13.9 | 29.0 | 14.8 | 13.6 | 14.2 | 29.2 | 14.8 |
| | Tent | 32.4 | 13.3 | 28.9 | 14.8 | 15.8 | 13.7 | 32.2 | 14.2 |
| | TPT | 34.9 | 12.8 | 30.5 | 14.7 | 16.5 | 12.5 | 33.7 | 13.4 |
| | DART | 30.1 | 13.2 | 27.4 | 14.8 | 13.4 | 13.8 | 33.5 | 14.3 |
| | IOUFilter | 28.6 | 15.5 | 25.5 | 16.2 | 12.7 | 13.5 | 31.4 | 14.1 |
| | C-TPT | 35.4 | 12.5 | 30.8 | 14.6 | 16.6 | 12.1 | 34.1 | 13.0 |
| | ZS-Norm | 36.0 | 12.3 | 31.2 | 14.5 | 16.6 | **11.9** | 35.2 | **12.7** |
| | Penalty | 36.2 | 12.4 | 31.5 | 14.7 | 16.8 | 12.0 | 35.5 | 12.9 |
| | SaLS | 36.3 | 12.6 | 31.4 | 14.7 | 16.7 | 12.1 | 35.3 | 13.1 |
| | O-TPT | 36.5 | 12.7 | 31.8 | 14.6 | 16.9 | 12.3 | 37.5 | 13.3 |
| | InsCal (Ours) | **37.1** | **10.6** | **33.2** | 14.5 | **20.8** | 12.2 | **38.5** | 13.2 |

Table 3: Class-wise AP with different components enabled. EM stands for entropy minimization. MS means using multiple source training. And CEM is short for calibrated entropy minimization. We show the comparison results on data set Night Clear.

| | | | | | AP | | | | | | | | mAP |
|---|---|---|---|---|---|---|---|---|---|---|---|---|---|
| EM | TPT | MS | TGIA | CEM | Bus | Bike | Car | Motor | Person | Rider | Truck | | All |
| ✓ | ✗ | ✗ | ✗ | ✗ | 31.8 | 30.6 | 32.5 | 33.7 | 34.6 | 34.2 | 32.8 | | 33.1 |
| ✓ | ✓ | ✗ | ✗ | ✗ | 32.6 | 31.8 | 33.8 | 35.4 | 35.8 | 35.5 | 33.8 | | 34.7 |
| ✓ | ✓ | ✓ | ✗ | ✗ | 33.5 | 34.4 | 35.1 | 35.7 | 36.7 | 37.8 | 35.1 | | 37.5 |
| ✓ | ✓ | ✓ | ✓ | ✗ | 34.6 | 35.0 | 36.2 | 36.7 | 37.8 | 38.0 | 35.0 | | 36.1 |
| ✓ | ✓ | ✓ | ✓ | ✓ | **36.2** | **37.2** | **37.7** | **38.5** | **39.6** | **40.8** | **37.9** | | **38.5** |

Table 4: Open-vocabulary object detection over Day Foggy, novel category is masked with gray.

| Method | Bus | Bike | Car | Motor | Person | Rider | Traffic Light | mAP | D-ECE% |
|---|---|---|---|---|---|---|---|---|---|
| FR | 28.1 | 29.7 | **49.7** | 26.3 | 33.2 | 35.5 | 19.8 | 32.0 | 14.7 |
| GDINO | 33.2 | 33.4 | 33.8 | 35.7 | 36.9 | 37.5 | 31.8 | 34.1 | 12.9 |
| TPT | 34.4 | 33.3 | 34.2 | 36.7 | 37.9 | 38.8 | 32.4 | 34.9 | 13.2 |
| C-TPT | 35.1 | 33.6 | 35.5 | 38.0 | 39.2 | 39.1 | 33.1 | 35.4 | 12.5 |
| ZS-Norm | 35.7 | 36.1 | 38.8 | 40.3 | 39.9 | 40.3 | 33.9 | 36.0 | 12.3 |
| Penalty | 36.0 | 36.4 | 38.8 | 40.6 | 40.3 | 40.5 | 33.8 | 36.2 | 12.4 |
| SaLS | 36.1 | 36.3 | 38.6 | 40.7 | 40.4 | 40.7 | 33.7 | 36.3 | 12.6 |
| O-TPT | 36.2 | 36.5 | 38.9 | 40.7 | 40.5 | **40.9** | **34.0** | 36.5 | 12.7 |
| InsCal (Ours) | **36.5** | **36.8** | 38.8 | **40.7** | **42.4** | 39.7 | 33.7 | **37.1** | **10.6** |

## 6 CONCLUSION

In this work, we present InsCal, a fully test-time adaptation (FTTA) solution for object detection. We investigate the miscalibration issues in entropy minimization within FTTA and propose extending Test-time Prompt Tuning (TPT) to a multi-source setting with text-guided feature augmentation. To address the miscalibration problem, we introduce a novel learning objective that assigns instance-specific weights. Experiments conducted on various cross-domain object detection datasets demonstrate that InsCal effectively reduces miscalibration. Further extensions would include multi-modal adaptation, opening up to other modalities like audio, video, or sensor data; and scalable multi-source integration with meta-learning or federated learning.

## 7 REPRODUCIBILITY STATEMENT

We have taken multiple steps to ensure the reproducibility of our work. A detailed description of our proposed method and training objectives is provided in Section 4 of the main paper. Additional implementation details, hyperparameter settings, and dataset information are included in Appendix C. To further facilitate reproducibility, we provide an anonymous link to the source code and scripts for training and evaluation in Appendix E. All datasets used in our experiments are publicly available, and their references are properly provided.

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

# Supplementary Material

# Appendix

## Table of Contents

## A  NOTATIONS

The notations and their corresponding descriptions used in the main paper are presented in Table 5.

Table 5: Notations

| Notation | Description |
|---|---|
| $\mathcal{L}_{cls}$ | classification loss of object detection |
| $\mathcal{L}_{reg}$ | regression loss of object detection |
| $\mathbf{x}$ | image |
| $y$ | ground truth class label |
| $\mathbf{b}$ | ground truth bounding box coordinates |
| $s_{conf}$ | confidence score |
| $\hat{y}$ | predicted class label |
| $\hat{\mathbf{b}}$ | predicted bounding box coordinates |
| $\hat{s}_{conf}$ | predicted confidence score |
| $\gamma$ | IoU threshold |
| ECE | expected calibration error |
| D-ECE | detection expected calibration error |
| $I(m)$ | the set of all samples in bin $m$ |
| $\text{prec}(m)$ | average precision in bin $m$ |
| $\text{conf}(m)$ | average confidence in bin $m$ |
| $\mathcal{D}_s$ | $s$-th source dataset |
| $\mathbf{x}_{test}$ | test image from the target dataset |
| $\text{ENC}_I^s$ | pre-trained $s$-th source image encoder |
| $\text{ENC}_T$ | pre-trained $s$-th text encoder |
| $S$ | number of source domains |
| $\mathbf{p}$ | learnable test-time prompt |
| $\theta$ | weight parameters for TGIA |
| $\mathcal{A}_\theta(\cdot)$ | TGIA augmentation |
| $\mathcal{A}(\cdot)$ | TGIA augmentation |
| $\text{src}_{sty}$ | source style text |
| $\text{tgt}_{sty}$ | target style text |
| $\mathbf{z}$ | image patch embedding |
| $\Delta I$ | difference between augmented image embedding and original image embedding |
| $\Delta T$ | difference between source style and target style |
| $\lambda$ | hyperparameter |
| $SIM(\cdot)$ | cosine similarity |
| $N$ | number of augmentations for each view |
| $\rho$ | cutoff percentile of augmented views |
| $y^{\text{1st}}$ | class with highest prediction normalized logit |
| $y^{\text{2nd}}$ | class with second highest prediction normalized logit |
| $p^{\text{1st}}$ | highest prediction normalized logit |
| $p^{\text{2nd}}$ | second highest prediction normalized logit |

## B  DISCUSSION OF UNSUPERVISED CALIBRATION

It is extremely challenging for unsupervised domain adaptation (UDA) models to provide calibrated results due to the lack of labels in the target domain and semantics shift between the source and target domains. Despite its significance, miscalibrated UDA remains largely under-explored. PseudoCal Hu et al. (2024) provides a post-hoc solution for miscalibrated UDA through inference-stage mixup synthesis, which aim to turn unsupervised UDA into supervised one. Specifically, it first generates a set of pseudo labeled target set by taking convex combinations of multiple pairs of real target samples and their pseudo labels. Then they perform supervised calibration such as temperature scaling Guo et al. (2017) based on the pseudo-labeled target set.

However, this pseudo calibration method requires access to unlabeled target dataset, making it not directly applicable to FTTA where only a single test image is available. Instead, we propose a unsupervised calibration method for FTTA by utilizing the predictive probability vector. Without requiring access to labeled target dataset, we design a instance-specific weight based on the divergence between the largest logit and the second largest logit to calibrate the pre-trained source model. Despite the limited resources, we have achieved an improvement of ECE on DWD dataset of 6.3%.

## C EXPERIMENTAL DETAILS

**Implementation details.** The default prompt fine-tuning step is set to 10. We increase tuning steps on Night Clear to 15 and decrease the steps on Day Foggy and Dusk Rainy to 5, base on the data domain similarity and difficulty. In bounding box prediction, we remove box of low maximum confidence, i.e. last 30%. We only average the logit prediction of those box with larger than 65% IoU. The training is conducted with 2 A100.

**Art Image Dataset** The Art Image dataset is proposed by Inoue et al. (2018), it includes three types of art images, 1k clipart with 20 classes, 2k watercolor with 6 classes, and 2k comic with 6 classes. We only evaluate the common classes in the three datasets, that is, bike, bird, car, cat, dog, and person.

**Comparison baselines** Our method is compared with the following methods: (1) without adaptation (w/o adpt): Faster RCNN Ren et al. (2015) (FR), GDINO Liu et al. (2024a); (2) Unsupervised domain adaptation (UDA): Universal Adaptation Network (UAN) You et al. (2019) and Calibrated Multiple Uncertainties (CMU) Fu et al. (2020), Domain Adaptive Faster RCNN (DAF) Chen et al. (2018), Multi-adversarial Faster RCNN (MAF) He & Zhang (2019), Asymmetric Triway Faster Rcnn (ATF) He & Zhang (2020), Hierarchical Transferability Calibration Network (HTCN) Chen et al. (2020), Strong Weak Domain Adaptation (SWDA) Saito et al. (2019), Augmented Feature Alignment Network (AFAN) Wang et al. (2021a), Channel-wise Alignment for Adaptive Object Detection (CAD) Yang et al. (2020), Partial Alignment Asymmetric Tri-way Faster RCNN (PAATF) He et al. (2021), Paradigm Teacher Multi-Adversarial Faster RCNN (PTMAF) He et al. (2023b), Implicit Domain-invariant Faster-RCNN (IDF) Lang et al. (2022), Sequence Feature Alignment (SFA) Wang et al. (2021b). Universal Scale-Aware Domain Adaptive Faster RCNN (USDAF) Shi et al. (2022), Confused and Disentangled Extraction (CODE) Shi et al. (2024). (3) Fully test-time adaptation (FTTA) method:Tent Wang et al. (2020), TPT Shu et al. (2022), DART Liu et al. (2024b), and IOUFilter Ruan & Tang (2024); (4)Calibrated test-time prompt-tuning methods: C-TPTYoon et al. (2024), ZS-NormMurugesan et al. (2024), PenaltyMurugesan et al. (2024), SaLSMurugesan et al. (2024), and O-TPTSharifdeen et al. (2025).

## D ADDITIONAL EXPERIMENT RESULTS

In this section, we present some additional experimental results and details including some ablation studies in Appendix D.1, some qualitative result for TGIA and its corresponding detection results in Appendix D.2, and the class-specific analysis for the DWD and Art Image datasets in Appendix D.3.

### D.1 ABLATION STUDIES

**Impact of Hyperparameter** $\alpha$**.** For Dusk Rainy and Night Rainy data sets, because of the difficulty due to the combination the rainy effect and less lighting and the inborn complexity of unsupervised calibration, the ECE improvement is marginal. For relative simpler Day Foggy dataset where we have better lighting conditions, we achieve prominent improvement through calibration balance hyperparameter fine-tuning. As shown in Figure 6, we tested different $\alpha$ values ranging from 0.1 to 1, and the D-ECE is minimized when $\alpha = 1$.

**Calibration metrics.** We have tested the calibration performance with other calibration metrics including ACE, and SCE for both the comparative baselines and our method. The results, as shown in evaluated on Day Foggy, show that our method outperforms the baselines across all metrics.

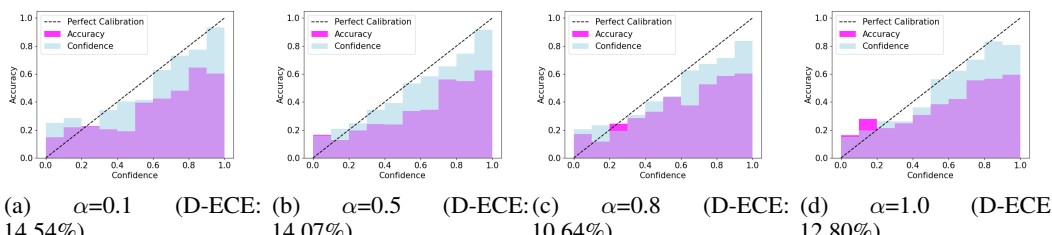

(a) $\alpha$=0.1 (D-ECE: 14.54%)
(b) $\alpha$=0.5 (D-ECE: 14.07%)
(c) $\alpha$=0.8 (D-ECE: 10.64%)
(d) $\alpha$=1.0 (D-ECE: 12.80%)

Figure 6: Comparison of TPT fine-tuned performance on Day Foggy from the DWD dataset Wu & Deng (2022) with different calibration balance factors $\alpha$ in training.

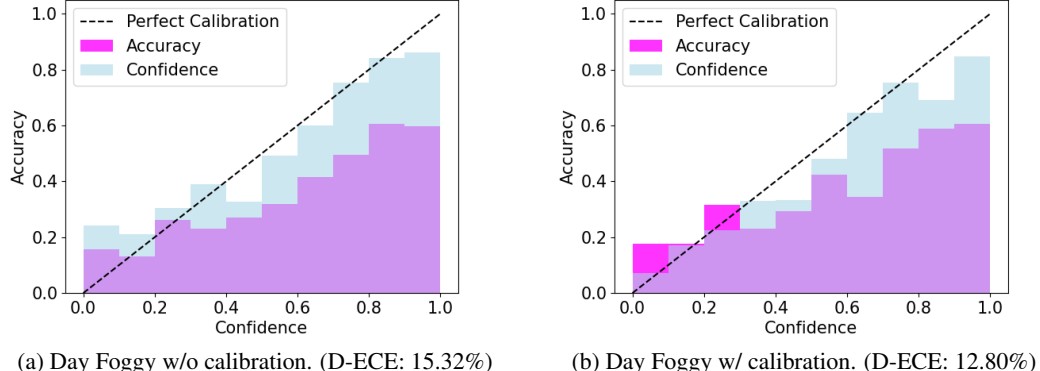

(a) Day Foggy w/o calibration. (D-ECE: 15.32%)  (b) Day Foggy w/ calibration. (D-ECE: 12.80%)

Figure 7: Comparison of TPT fine-tuned on Day Foggy from the DWD dataset Wu & Deng (2022) w/ and w/o calibration loss in training.

Table 6: Calibration performance by ACE and SCE.

| Methods | **ACE** (%) | **SCE** (%) |
|---|---|---|
| SED | 26.2 | 25.5 |
| MSMT | 27.5 | 26.8 |
| MixUp | 27.9 | 27.2 |
| HCL | 27.8 | 27.1 |
| IRG | 30.5 | 29.7 |
| GDINO | 22.0 | 21.2 |
| Tent | 23.1 | 22.2 |
| TPT | 23.2 | 22.0 |
| C-TPT | 22.9 | 21.6 |
| ZS-Norm | 23.0 | 21.6 |
| Penalty | 23.1 | 21.7 |
| SaLS | 23.2 | 21.8 |
| O-TPT | 23.4 | 21.9 |
| DART | 20.9 | 19.8 |
| IOUFilter | 34.3 | 33.5 |
| **InsCal(Ours)** | **18.5** | **17.8** |

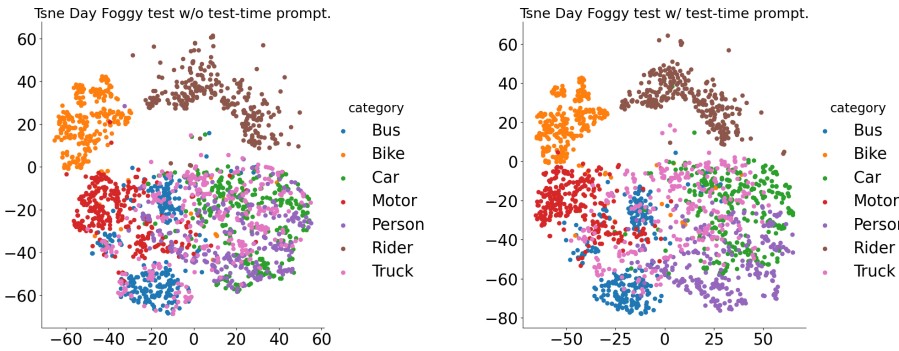

Figure 8: T-SNE results w/ and w/o calibrated test-time prompt.

**Interpretability.** To demonstrate interpretability, we present T-SNE results comparing calibrated test-time prompt tuning (CTPT) and the uncalibrated version in Figure 8. In the left figure, Truck, Car, and Person are blended, whereas in the right figure, they are clearly separated, highlighting the effectiveness of CTPT.

**Aggregation of multiple source models.** In this ablation, we study the aggregation of multiple pre-trained source models. Results show that directly aggregate information from multiple source models is not always useful, sometimes may even have a negative impact on the adaptation to the target domain. As shown in Figure 9, we use different combinations of source models, and adapt to the unseen target models on DWD dataset. The results on Night Clear show that using Night Rainy achieve best performance, while incorporating other domains such as Dusk Rainy or Day Foggy harm the performance. Similar results can be observed from Day Foggy and Dusk Rainy domains. For Night Rainy, incorporating sources such as Dusk Rainy, Day Foggy improve the performance. The reason is that, Night Rainy show similar semantic with Night Clear. For example, they are both night images. Using Night Rainy as source will lead to a good performance on Night Clear dataset. However, there is a huge domain gap between other domains and Night Clear, including Dusk Rainy and Day Foggy. Incorporation of these domain as sources will induce negative transfer, further degrade the transferring from Night Rainy to Night Clear.

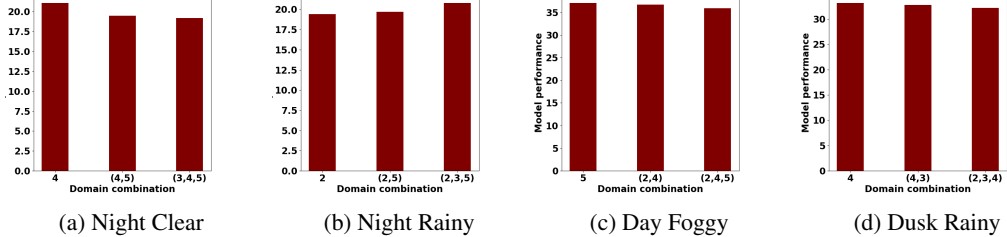

| (a) Night Clear | (b) Night Rainy | (c) Day Foggy | (d) Dusk Rainy |

Figure 9: mAP results on DWD domains 9a-9d. In each domain we use different combination of source models as multi-source models, each number stands for a specific domain: 1-Day Clear, 2-Night Clear, 3-Day Foggy, 4-Night Rainy, and 5-Dusk Rainy.

**Computational Complexity Analysis.** Table 7 presents the time and memory analysis of our method compared to baseline approaches. InsCal incurs a slight increase in inference time and parameter size compared to TPT while outperforming other FTTA methods.

Table 7: Time and Memory Complexity

|                       | Tent   | TPT  | CTPT | O-TPT | DART | IOUFilter | InsCal |
|-----------------------|--------|------|------|-------|------|-----------|--------|
| Inference Time (FPS)  | **0.18** | 0.25 | 0.25 | 0.24  | 0.29 | 0.28      | 0.24   |
| Parameter Size (M)    | **75**   | 84   | 85   | 85    | 92   | 93        | 85     |

## D.2 Qualitative Results of TGIA

We present some augmented images using TGIA on the art images dataset in Figure 10. In Table 8, we presented the text-based style description for the corresponding source domain. We formulate the source domain style description by simply prepending the template "A photo of" to the dataset name, which requires no extra information and no access to the source domain images. As shown in Figure 10, TGIA manages to transfer the source domain styles to the target images while preserving the details of the original content. The corresponding detection results are shown in Figure 11, where we present the detection performance using entropy minimization (EM), TPT, multiple source (MS) and our method InsCal. The detection results show that EM and TPT both fail to detect multiple people in the figure. And utilizing multiple source and InsCal improve the detection performance.

Table 8: Source domains and its corresponding descriptions.

| Source | Description |
|---|---|
| Clipart | A photo of clipart |
| Comic | A photo of comic |
| Watercolor | A photo of watercolor |
| Day Foggy | A photo of foggy day |

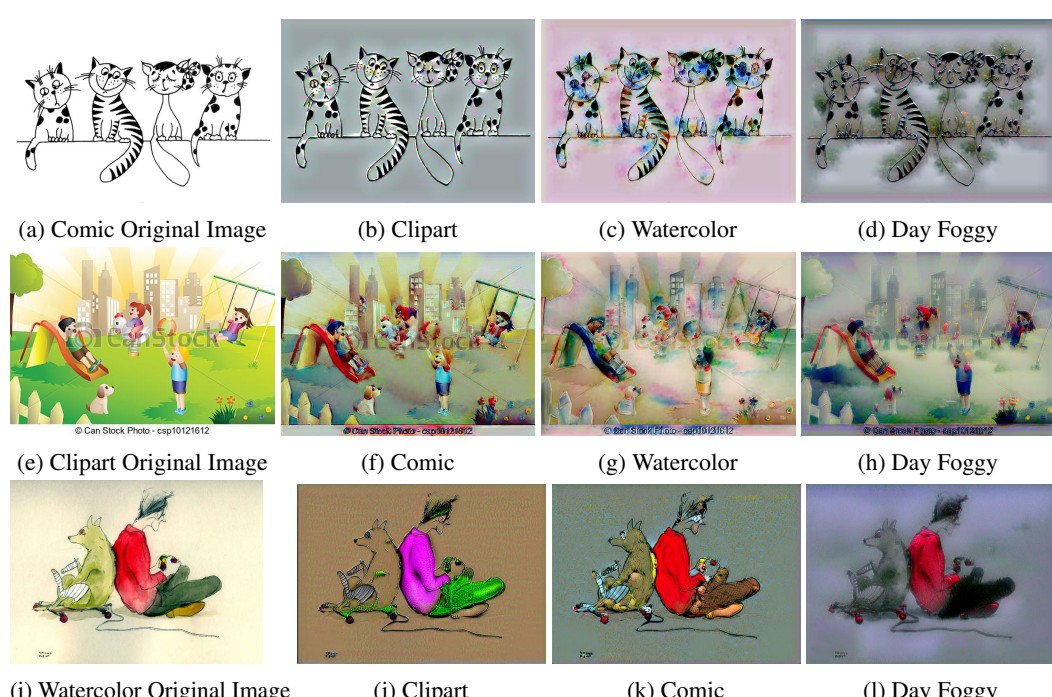

(a) Comic Original Image (b) Clipart (c) Watercolor (d) Day Foggy

(e) Clipart Original Image (f) Comic (g) Watercolor (h) Day Foggy

(i) Watercolor Original Image (j) Clipart (k) Comic (l) Day Foggy

Figure 10: Images from Comic (a), Clipart (e) and Watercolor (i) are transfer to clipart, comic, watercolor and day foggy style with source description in Table 8, respectively.

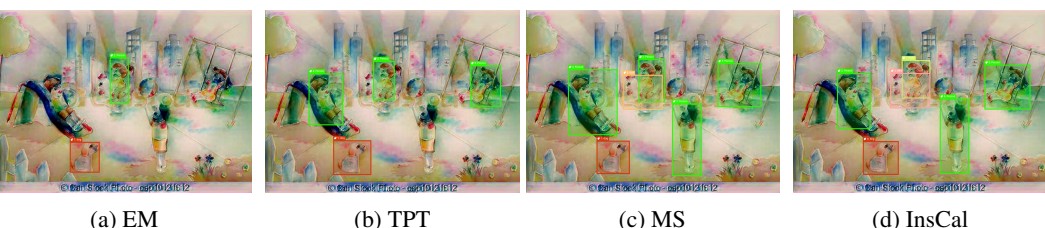

(a) EM (b) TPT (c) MS (d) InsCal

Figure 11: Ablative model components cross-domain detection results on Art Watercolor datasets using different comparative baselines.

## D.3 CLASS-SPECIFIC RESULTS ANALYSIS

In this section, we provide the class-specific results analysis for each domain in DWD and the Art Image datasets. Since some UDA and SFDA methods mentioned in Table 2 do not provide the details of class-specific results, we remove them from the per-class analysis in this section.

**Day Foggy.** In Table 9, we observe that InsCal has outperform all the FTTA methods in every category. Some categories such as Motor, Person, and Truck achieve the highest performance across UDA, SFDA and FTTA.

**Dusk Rainy.** As shown in Table 10, in UDA, CLIPAug performs better than other methods in every category. For SFDA, MixUp performans consistently better than other methods. In FTTA, our method outperforms other methods in every category. In addition, our methods performs better than CLIPAug and MixUp in most of the categories, showing a better overall mAP across UDA, SFDA and FTTA.

**Night Rainy.** As presented in Table 11, CLIPAug provides consistently better performance than other methods, resulting a 18.7 mAP, outperforming the rest of the UDA methods. The lack of lighting (Night) and the raining effect makes the overall detection on night rainy domain very difficult. In SFDA, MSMT performs consistently better than the others. The utilization of multiple source models and the unlabeled target images lead to the stable performance of 29.1 mAP. In FTTA, our methods outperform the rest of the methods in every category, leading to the highest mAP, which is better than UDA but inferior to SFDA due to the lack of utilization of the unlabeled target images.

**Night Clear.** In Table 12, we present the class-wise results for Night Clear. In UDA, SDGOD and CLIPAug have the best performance in different categories, and their mAP are very close. In SFDA, MSMT performs consistently better than the others. In FTTA, our method performs better than others across all the categories. However, the base model GDINO has difficulty adapting to the night domain, making the overall performance consistently worse than UDA and SFDA.

Table 9: Class-wise mAP on Day Foggy domain of DWD dataset.

| Class Names | Method | Bus | Bike | Car | Motor | Person | Rider | Truck | mAP |
|---|---|---|---|---|---|---|---|---|---|
| | FR | 28.1 | 29.7 | 49.7 | 26.3 | 33.2 | 35.5 | 21.5 | 32.0 |
| **UDA** | SDGOD | 32.9 | 28.0 | 48.8 | 29.8 | 32.5 | 38.2 | 24.1 | 33.5 |
| | CLIPAug | **36.1** | **34.3** | **58.0** | **33.1** | **39.0** | **43.9** | **25.1** | **38.5** |
| | SED | 28.4 | 29.1 | 28.5 | 24.1 | 33.9 | 30.4 | 32.7 | 29.4 |
| | MSMT | **35.4** | **37.9** | **40.2** | **39.2** | 31.5 | 33.4 | 32.9 | **36.8** |
| **SFDA** | MixUp | 33.2 | 32.4 | 33.5 | 26.8 | 29.1 | 35.5 | 33.2 | 31.5 |
| | HCL | 32.5 | 31.3 | 32.1 | 25.9 | 28.0 | 34.2 | 31.8 | 30.2 |
| | IRG | 33.8 | 33.9 | 34.2 | 36.8 | **37.5** | **38.9** | **34.8** | 35.2 |
| | GDINO | 33.2 | 33.4 | 33.8 | 35.7 | 36.9 | 37.5 | 33.5 | 34.1 |
| | Tent | 31.0 | 31.3 | 31.9 | 33.7 | 34.9 | 35.7 | 31.6 | 32.4 |
| | TPT | 34.4 | 33.3 | 34.2 | 36.7 | 37.9 | 38.8 | 34.7 | 34.9 |
| | C-TPT | 35.1 | 33.6 | 35.5 | 38.0 | 39.2 | 39.1 | 33.1 | 35.4 |
| | ZS-Norm | 35.7 | 36.1 | 38.8 | 40.3 | 39.9 | 40.3 | 33.9 | 36.0 |
| **FTTA** | Penalty | 36.0 | 36.4 | 38.8 | 40.6 | 40.3 | 40.5 | 33.8 | 36.2 |
| | SaLS | 36.1 | 36.3 | 38.6 | 40.7 | 40.4 | 40.7 | 33.7 | 36.3 |
| | O-TPT | 36.2 | 36.5 | 38.9 | 40.7 | 40.5 | **40.9** | **34.0** | 36.5 |
| | DART | 28.8 | 27.2 | 28.9 | 31.4 | 32.6 | 32.9 | 29.2 | 30.1 |
| | IOUFilter | 35.9 | 24.8 | 25.6 | 28.7 | 30.9 | 30.5 | 27.5 | 28.6 |
| | InsCal | **36.5** | **36.8** | **38.8** | **40.7** | **42.4** | 39.7 | 33.7 | **37.1** |

**Class-specific results on Comic** In Table 13, we present the class-wise AP for Comic. As observed, InsCal performs consistently better across different classes, resulting the highest mAP. GDINO performs better then FR across all categories, thanks to the larger pre-trained datasets. UDA methods perform better on some classes such as bike, car or dog, but InsCal has a better mAP due to the advantages over the rest of the classes. By utilizing multiple source models and calibration, InsCal achieve comparable performance with UDA methods.

**Class-specific results on Watercolor** In Table 14, we present the class-wise results for Watercolor dataset. Similar results can be observed from Table 14 for the Watercolor dataset. Showing the consist trend of our method and the comparison between InsCal and other methods. TPT performs well in cat category, but InsCal still performs better on the rest classes, resulting a higher mAP.

Table 10: Class-wise mAP on Dusk Rainy domain of DWD dataset.

| Type | Method | Bus | Bike | Car | Motor | Person | Rider | Truck | mAP |
|------|--------|-----|------|-----|-------|--------|-------|-------|-----|
| UDA | FR | 28.5 | 20.3 | 58.2 | 6.5 | 23.4 | 11.3 | 33.9 | 26.0 |
| | SDGOD | 37.1 | 19.6 | 50.9 | 13.4 | 19.7 | 16.3 | 40.7 | 28.2 |
| | CLIPAug | **27.8** | **28.8** | 28.7 | **27.2** | **27.9** | **28.3** | 28.4 | 28.2 |
| SFDA | SED | 13.5 | 16.3 | 16.2 | 16.5 | 14.4 | 15.3 | 14.9 | 15.4 |
| | MSMT | **34.1** | **33.6** | **31.9** | **31.4** | **31.7** | **32.3** | 32.3 | **32.0** |
| | MixUp | 29.2 | 28.9 | 28.8 | 29.8 | 31.0 | 31.2 | **32.9** | 30.8 |
| | HCL | 26.7 | 27.2 | 27.4 | 27.2 | 25.9 | 25.3 | 26.5 | 26.9 |
| | IRG | 31.2 | 28.4 | 28.8 | 30.9 | 31.5 | 28.3 | 29.7 | 30.5 |
| FTTA | GDINO | 27.1 | 25.8 | 29.2 | 29.4 | 30.1 | 31.5 | 28.5 | 29.0 |
| | Tent | 26.8 | 25.9 | 29.5 | 29.2 | 27.3 | 30.4 | 28.7 | 28.9 |
| | TPT | 28.8 | 27.6 | 30.8 | 31.2 | 31.5 | 32.9 | 30.1 | 30.5 |
| | C-TPT | 29.2 | 27.8 | 30.9 | 31.4 | 31.6 | 33.4 | 30.6 | 30.8 |
| | ZS-Norm | 29.8 | 28.0 | 31.2 | 31.6 | 31.9 | 33.7 | 31.2 | 31.2 |
| | Penalty | 30.0 | 28.2 | 31.6 | 32.0 | 32.4 | 34.1 | 31.4 | 31.5 |
| | SaLS | 29.9 | 28.3 | 31.8 | 31.6 | 32.5 | 33.8 | 31.2 | 31.4 |
| | O-TPT | 30.4 | 28.8 | 32.3 | 31.9 | 32.7 | 34.3 | 31.5 | 31.8 |
| | DART | 25.4 | 24.7 | 28.3 | 28.1 | 26.4 | 29.0 | 27.1 | 27.4 |
| | IOUFilter | 23.5 | 22.9 | 26.5 | 26.4 | 24.8 | 27.2 | 24.9 | 25.5 |
| | InsCal | **32.9** | **31.7** | **32.5** | **34.3** | **35.6** | **36.5** | **32.8** | **33.2** |

Table 11: Class-wise mAP on Night Rainy domain of DWD dataset.

| Type | Method | Bus | Bike | Car | Motor | Person | Rider | Truck | mAP |
|------|--------|-----|------|-----|-------|--------|-------|-------|-----|
| UDA | FR | 16.8 | 6.9 | 26.3 | 0.6 | 11.6 | 9.4 | 15.4 | 12.4 |
| | SDGOD | 24.4 | 11.6 | 29.5 | **9.8** | 10.5 | **11.4** | 19.2 | 16.6 |
| | CLIPAug | **28.6** | **12.1** | **36.1** | 9.2 | **12.3** | 9.6 | **22.9** | **18.7** |
| SFDA | SED | 15.8 | 14.5 | 14.2 | 18.6 | 6.9 | 16.5 | 18.8 | 15.1 |
| | MSMT | **16.9** | **16.8** | **16.4** | **16.6** | **16.2** | **16.8** | **16.7** | **16.5** |
| | MixUp | 15.6 | 15.2 | 15.4 | 15.8 | 15.7 | 15.2 | 15.3 | 15.5 |
| | HCL | 15.2 | 14.8 | 15.0 | 15.5 | 15.6 | 15.7 | 15.4 | 15.3 |
| | IRG | 15.5 | 15.5 | 15.6 | 16.1 | 16.3 | 16.5 | 15.7 | 15.8 |
| FTTA | GDINO | 12.5 | 12.3 | 13.9 | 14.2 | 14.5 | 14.8 | 13.2 | 13.6 |
| | Tent | 13.7 | 13.4 | 14.5 | 16.3 | 17.1 | 17.2 | 15.0 | 15.8 |
| | TPT | 14.5 | 14.1 | 15.8 | 17.2 | 18.0 | 17.8 | 15.8 | 16.5 |
| | C-TPT | 14.6 | 14.4 | 15.9 | 17.0 | 17.9 | 17.9 | 16.0 | 16.6 |
| | ZS-Norm | 14.8 | 14.3 | 16.0 | 17.2 | 17.8 | 17.8 | 16.1 | 16.6 |
| | Penalty | 14.9 | 14.6 | 16.2 | 17.3 | 17.9 | 18.1 | 16.4 | 16.8 |
| | SaLS | 14.8 | 14.4 | 16.3 | 17.4 | 17.8 | 18.0 | 16.2 | 16.7 |
| | O-TPT | 14.7 | 14.3 | 16.2 | 17.7 | 17.9 | 18.2 | 16.5 | 16.9 |
| | DART | 11.2 | 11.0 | 12.5 | 14.8 | 15.9 | 13.7 | 12.9 | 13.4 |
| | IOUFilter | 10.8 | 10.5 | 12.0 | 14.2 | 15.5 | 13.1 | 12.4 | 12.7 |
| | InsCal | **21.8** | **22.2** | **21.8** | **22.7** | **25.8** | **23.5** | **20.8** | **20.8** |

Table 12: Class-wise mAP on Night Clear domain of DWD dataset.

| Type | Method | Bus | Bike | Car | Motor | Person | Rider | Truck | mAP |
|------|--------|-----|------|-----|-------|--------|-------|-------|-----|
| UDA | FR | 34.7 | 32 | 56.6 | 13.6 | 36.8 | 27.6 | 38.6 | 34.4 |
| | SDGOD | **40.6** | **35.1** | 45.7 | 19.7 | 34.7 | **32.1** | **43.4** | 36.6 |
| | CLIPAug | 37.7 | 34.3 | **48.0** | **29.2** | **37.6** | 28.5 | 42.9 | **36.9** |
| SFDA | SED | 31.9 | 34.5 | 33.8 | 31.2 | 32.5 | 34.9 | 33.7 | 33.4 |
| | MSMT | **38.2** | **35.8** | **39.2** | **39.0** | **43.2** | **38.1** | **37.0** | **37.7** |
| | MixUp | 35.8 | 34.2 | 34.5 | 34.6 | 36.0 | 36.2 | 34.5 | 35.0 |
| | HCL | 29.4 | 31.9 | 31.2 | 29.5 | 29.9 | 32.4 | 31.2 | 30.8 |
| | IRG | 37.3 | 35.8 | 36.4 | 36.5 | 37.8 | 37.9 | 36.1 | 36.7 |
| FTTA | GDINO | 27.6 | 26.5 | 28.8 | 29.9 | 30.5 | 30.4 | 28.5 | 29.2 |
| | Tent | 29.6 | 28.5 | 30.8 | 31.9 | 32.5 | 32.4 | 30.5 | 32.2 |
| | TPT | 33.6 | 32.8 | 32.8 | 34.4 | 34.8 | 34.5 | 32.8 | 33.7 |
| | C-TPT | 33.7 | 33.0 | 33.1 | 34.9 | 35.4 | 34.8 | 33.3 | 34.1 |
| | ZS-Norm | 34.7 | 34.1 | 34.3 | 36.2 | 36.3 | 35.7 | 34.5 | 35.2 |
| | Penalty | 34.9 | 34.3 | 34.8 | 36.4 | 36.4 | 36.1 | 34.8 | 35.5 |
| | SaLS | 34.8 | 34.1 | 34.5 | 36.2 | 36.0 | 36.0 | 34.5 | 35.3 |
| | O-TPT | 35.6 | 35.5 | 35.4 | 37.3 | 37.4 | 37.2 | 35.6 | 37.5 |
| | DART | 31.4 | 30.7 | 32.5 | 34.5 | 34.9 | 34.8 | 33.0 | 33.5 |
| | IOUFilter | 29.2 | 28.8 | 30.3 | 32.4 | 33.0 | 32.6 | 31.1 | 31.4 |
| | InsCal | **36.3** | **37.1** | **37.7** | **38.8** | **39.5** | **40.8** | **37.9** | **38.5** |

Table 13: Class-specific AP on Comic Dataset.

| Category | Methods | bike | bird | car | cat | dog | person | mAP |
|----------|---------|------|------|-----|-----|-----|--------|-----|
| w/o adpt | FR | 39.6 | 11.3 | 30.4 | 12.9 | 15.4 | 40.3 | 25.0 |
| | GDINO | **40.7** | **12.4** | **31.4** | **13.8** | **16.2** | **50.2** | **25.9** |
| UDA | UAN | 41.0 | 16.0 | 29.1 | 8.6 | 14.4 | 43.8 | 25.5 |
| | CMU | 36.8 | 17.8 | 24.5 | 18.3 | 28.9 | 54.5 | 30.1 |
| | DAF | 32.1 | 21.3 | 26.4 | 12.5 | **31.1** | 46.2 | 28.3 |
| | MAF | **43.1** | 17.5 | 24.2 | 19.4 | 22.4 | 49.1 | 29.3 |
| | HTCN | 30.0 | 13.9 | 27.7 | 7.5 | 26.1 | 38.4 | 24.0 |
| | CAD | 39.1 | 24.8 | 25.8 | 11.0 | 22.0 | 49.9 | 28.8 |
| | IDF | 19.9 | 20.5 | 25.8 | 15.0 | 22.8 | 44.6 | 24.8 |
| | USDAF | 39.8 | 15.9 | **38.6** | 18.1 | 26.6 | 56.5 | 32.6 |
| | CODE | 40.2 | **26.9** | 29.7 | **19.5** | 26.6 | **59.8** | **33.8** |
| FTTA | Tent | - | - | - | - | - | - | 25.5 |
| | TPT | 40.8 | 12.6 | 31.5 | 13.7 | 16.4 | 50.1 | 25.9 |
| | IOUFilter | - | - | - | - | - | - | 20.2 |
| | InsCal | **40.7** | **27.4** | **30.2** | **20.1** | **27.3** | **60.3** | **34.3** |

Table 14: Class-specific AP on Watercolor

| Category | Methods | bike | bird | car | cat | dog | person | mAP |
|---|---|---|---|---|---|---|---|---|
| w/o adpt | FR | 82.4 | 51.7 | 48.4 | 39.9 | 30.7 | 59.2 | 52.0 |
| | GDINO | **83.1** | **52.5** | **49.4** | **40.9** | **31.7** | **59.9** | **52.8** |
| UDA | UAN | 78.0 | 53.6 | 50.4 | 36.4 | 35.8 | 65.6 | 53.3 |
| | CMU | 82.0 | 53.9 | 48.6 | 39.6 | 33.1 | 66.0 | 53.9 |
| | DAF | 73.4 | 51.9 | 43.1 | 35.6 | 28.8 | 63.1 | 49.3 |
| | MAF | 70.4 | 50.3 | 44.3 | 36.7 | 30.6 | 62.9 | 49.2 |
| | HTCN | 74.1 | 49.8 | **51.9** | 35.3 | **35.3** | 66.0 | 52.1 |
| | CAD | 82.3 | 52.3 | 49.3 | 38.1 | 32.0 | 62.6 | 52.8 |
| | IDF | 81.4 | 54.9 | 46.7 | 36.6 | 29.1 | 66.0 | 52.5 |
| | USDAF | 86.5 | 54.1 | 50.0 | **43.0** | 34.0 | 63.2 | 55.2 |
| | CODE | **87.9** | **55.3** | 50.7 | 38.9 | 34.7 | **67.5** | **55.8** |
| FTTA | Tent | - | - | - | - | - | - | 52.5 |
| | TPT | 83.2 | 52.6 | 49.5 | **41.1** | 31.9 | 60.2 | 53.0 |
| | IOUFilter | - | - | - | - | - | - | 35.8 |
| | InsCal | **88.38** | **55.7** | **51.3** | 39.45 | **35.3** | **68.1** | **56.3** |

**Class-specific results on Clipart** In Table 15, we can draw similar conclusions of the per-class analysis for Clipart dataset as the previous Comic and Watercolor datasets, where InsCal provides consistent and stable performance.

Table 15: Class-specific AP on Clipart

| Category | Methods | bike | bird | car | cat | dog | person | mAP |
|---|---|---|---|---|---|---|---|---|
| w/o adpt | FR | - | - | 34.7 | 5.1 | 8.3 | 49.6 | 29.8 |
| | GDINO | 56.9 | 18.6 | **34.8** | **5.2** | **8.4** | **50.5** | **30.5** |
| UDA | UAN | - | - | 31.5 | 8.6 | 2.4 | 42.8 | 30.3 |
| | CMU | - | - | 34.7 | 9.2 | 7.6 | 55.7 | 32.1 |
| | DAF | - | - | 35.9 | 2.3 | 4.2 | 59.4 | 31.3 |
| | MAF | - | - | 32.3 | 11.0 | 6.7 | 52.7 | 32.2 |
| | HTCN | - | - | 32.8 | 11.3 | **10.5** | 57.9 | 34.7 |
| | CAD | - | - | 35.9 | 9.8 | 4.7 | 56.1 | 34.2 |
| | IDF | - | - | 37.3 | 16.7 | 3.7 | 52.6 | 32.7 |
| | USDAF | - | - | 36.4 | 17.7 | 10.3 | **62.5** | 38.4 |
| | CODE | - | - | **37.7** | **18.4** | 8.4 | 61.7 | **39.4** |
| FTTA | Tent | - | - | - | - | - | - | 30.3 |
| | TPT | 60.1 | 18.3 | 35.5 | 10.1 | 8.4 | 50.7 | 30.6 |
| | IOUFilter | - | - | - | - | - | - | 29.6 |
| | InsCal | **69.4** | **28.2** | **38.1** | **18.9** | **10.0** | **62.4** | **39.9** |

# E SOURCE CODE

For source code, please refer to `https://anonymous.4open.science/r/InsCal-6602/README.md`.

# F LLM USAGE STATEMENT

Large Language Models (LLMs) were used solely to aid in polishing the writing and improving the clarity of exposition. No part of the research ideation, experimental design, implementation, or analysis relied on LLMs. The authors take full responsibility for the content of this paper.

