# OpenReview forum: "InsCal: Calibrated Multi-Source Fully Test-Time Prompt Tuning for Object Detection"
_ICLR.cc/2026/Conference — ICLR 2026 Conference Withdrawn Submission_

### Official Review · Reviewer_HhSq · 2025-10-30

**Soundness:** 2
**Presentation:** 2
**Contribution:** 2
**Rating:** 2
**Confidence:** 5

**Summary:**

This paper focuses on test-time domain adaptation task, aiming to transfer knowledge learned from the labelled source domain to an unlabeled target domain, where the target samples are provided one at a time. To solve this, the paper proposes  InsCal , a test-time adaptation (TTA) approach for text-driven object detection that tackles the problem of miscalibration during adaptation. Experiments show that InsCal outperforms existing methods on cross-domain datasets by significantly reducing D-ECE (detection expected calibration error) and improving mAP (mean average precision).

**Strengths:**

* The paper is easy to follow.
* The test-time adaptation is an interesting problem, and it is even more important when trying to deploy VLM to real-world applications.

**Weaknesses:**

1. Limited novelty. While InsCal integrates multiple source models and improves TPT, the approach heavily relies on existing methods such as text-guided generalization and entropy minimization. Its contribution lies more in integrating these techniques. For example, the proposed TGIA strictly follows CLIPtheGap[1], and MSTPT seems like a direct extension of TPT.
2. It appears that TGIA's performance depends on the accuracy of the textual description of the domain. If the domain is no longer a difference in weather conditions or image style, but rather an attribute that is difficult to describe in text, TGIA may not be effective, and the paper lacks similar analysis.
3. The necessity of task definition. Is it necessary to define multi-source TPT? Throughout the process, the method's operation on the domain is only reflected in using different prompts to guide image enhancement, which is a method already applied in existing work and can hardly be considered a contribution of this paper.
4. Limited performance improvements and unfair comparisons. Since the baseline uses GDINO, and its zero-shot performance already surpasses some state-of-the-art (SOTA) works, direct comparisons with other works are unfair. Specifically, results of fine-tuning GDINO using TPT or O-TPT should be reported, and only the performance differences between the proposed method and them can be considered as an absolute improvement.

[1]Vidit V, Engilberge M, Salzmann M. Clip the gap: A single domain generalization approach for object detection. CVPR 2023

**Questions:**

* Please refer to Weakness.
* Why the provided UDA and SFDA methods are NOT SOTA? We all know that some existing UDA and SFDA work has achieved better performance.

---

### Official Review · Reviewer_yxVi · 2025-10-31

**Soundness:** 2
**Presentation:** 2
**Contribution:** 2
**Rating:** 4
**Confidence:** 4

**Summary:**

In this paper, the authors propose InsCal to enhance the performance of multi-domain object detection. Utilizing text-driven style transfer strategy to align features from different domains, InsCal mitigates miscalibration caused by domain shifts.

**Strengths:**

1.	The authors propose a test-time adaptation method, InsCal, for multi-source object detection setting.
2.	InsCal mitigates the miscalibration by refine instance-level entropy minization objective.

**Weaknesses:**

1.	The performance improvement achieved by InsCal appears to be limited. Moreover, although UDA and SFDA methods have access to target-domain data, FTTA methods are allowed to directly access test samples. Therefore, the statement in Lines 375–376 is not entirely rigorous.
2.	The overall design of the proposed approach lacks originality. The main framework of InsCal heavily relies on TPT, and the text-guided feature augmentation strategy has already been explored in several prior works [1,2].
3.	The benchmarks used for evaluation are relatively small. It would strengthen the paper to include experiments on larger-scale benchmarks to further validate the generalization capability of the proposed method.
4.   Typos: There are two consecutive commas in Line 269.

[1] PromptStyler: Prompt-driven Style Generation for Source-free Domain Generalization (ICCV23)

[2] Using Language to Extend to Unseen Domains (ICLR23)

**Questions:**

1.	In Lines 252–254, it is unclear why the magnitude scaling factor is designed to match the transformation magnitude of the desired text-guided shift.
2.	The InsCal method requires performing N augmentations for each of the S source domains. Considering that the task is TTA, what are the associated computational and time costs of InsCal? In addition, key hyperparameters such as N and ρ should be explicitly described and discussed in the main text.

---

### Official Review · Reviewer_bMFe · 2025-11-01

**Soundness:** 2
**Presentation:** 2
**Contribution:** 2
**Rating:** 2
**Confidence:** 3

**Summary:**

This paper, InsCal, proposes a new framework for Fully Test-Time Adaptation (FTTA) of text-driven object detectors. The authors identify that standard Test-Time Prompt Tuning (TPT), which relies on entropy minimization, suffers from miscalibration and overconfidence when faced with domain shifts. To address this, InsCal introduces three components: (1) a multi-source TPT paradigm that aggregates knowledge from multiple source-specific models; (2) a Text-Guide Image Augmentation (TGIA) module that uses text-driven style transfer to reduce the domain gap; and (3) a novel "instance-specific calibrated" entropy loss (InsCal) that re-weights the entropy minimization objective based on the confidence margin between the top two predictions. The authors claim their method significantly reduces calibration error and improves mAP on cross-domain object detection tasks.

**Strengths:**

The paper addresses an important and practical problem: the application of test-time adaptation to open-vocabulary object detectors, which is a relatively under-explored area. The authors correctly identify a key failure mode of standard TPT: entropy minimization can exacerbate miscalibration by encouraging overconfident, incorrect predictions, especially under domain shift.

**Weaknesses:**

1. The paper's claim of state-of-the-art performance is based on a comparison that appears to provide the proposed method with more resources. InsCal is presented as "multi-source" and uses four source models (fine-tuned on Day Clear + three other domains, per Sec 4.2 and Table 2 caption). In contrast, the competing FTTA baselines (TPT, O-TPT, etc.) are treated as single-source (using only Day Clear, per Table 2 caption). This discrepancy makes it difficult to isolate the performance gains of the proposed calibration method from the benefits of simply using more source models and data. The ablation in Fig. 9, which shows that adding sources can sometimes harm performance, further complicates this.

2. The framework's requirements seem to challenge the standard constraints of "Fully Test-Time Adaptation" (FTTA). The method appears to require fine-tuning the base GDINO model on multiple, diverse source datasets to create the "source models." This represents a significant setup cost compared to the more common FTTA premise of adapting a single, off-the-shelf pre-trained model. The Text-Guide Image Augmentation (Sec 4.1) requires a "short text description of its style domain" for both the source and the target (e.g., "A photo of foggy day"). In a typical FTTA scenario, the target image arrives without a priori knowledge of its domain. Requiring a manually-provided text prompt for the target domain's style could be seen as a form of supervision, which may not align with the "fully test-time" setting.

3. The implementation details for the Text-Guide Image Augmentation (TGIA) module are unclear. This component is defined by an optimization process (Eq. 4) to train the augmentation network $\mathcal{A}_{\theta}$. The paper does not specify when this optimization occurs. If it is run for every single test image, it would introduce a significant computational overhead (an optimization within the adaptation loop) that is not analyzed in the paper's complexity discussion. Clarification on this process is needed.

4. Several figures and captions are confusing or contain errors. (discussed in the questions below).

**Questions:**

1. The literature review appears to be incomplete. Many recent papers are not discussed [1-6].

2. On Figure 1(a): Why is the zero-shot performance compared to a "Fine-Tune" baseline? Since FTTA is an unsupervised method that adapts to unlabeled test data, what is the relevance of a fully supervised fine-tuned model to motivate this work?

3. On Figure 3: In the diagram for InsCal, "Source Model 3" is listed twice.

4. The caption for Figure 1(a) is placed awkwardly close to the caption for 1(b), making it difficult to read.

[1] Ra-tta: Retrieval-augmented test-time adaptation for vision-language models, ICLR2025

[2] Bayesian test-time adaptation for vision-language models, CVPR2025

[3] Efficient and context-aware label propagation for zero-/few-shot training-free adaptation of vision-language model, ICLR2025

[4] DPCore: Dynamic prompt coreset for continual test-time adaptation, ICML2025

[5] DynaPrompt: Dynamic Test-Time Prompt Tuning, ICLR2025

[6] Test-Time Model Adaptation with Only Forward Passes, ICML2024

---

### Note · Authors · 2025-11-14

I have read and agree with the venue's withdrawal policy on behalf of myself and my co-authors.